# Insights into the Applications of Natural Fibers to Metal Separation from Aqueous Solutions

**DOI:** 10.3390/polym15092178

**Published:** 2023-05-03

**Authors:** Lavinia Tofan

**Affiliations:** Department of Environmental Engineering and Management, “Cristofor Simionescu” Faculty of Chemical Engineering and Environmental Protection, “Gheorghe Asachi” Technical University of Iasi, 73 Prof.Dr. D. Mangeron Blvd., 700050 Iasi, Romania; lavinia_tofan@yahoo.com

**Keywords:** natural lignocellulosic fibers, natural protein fibers, biosorption, metals, removal, recycling

## Abstract

There is a wide range of renewable materials with attractive prospects for the development of green technologies for the removal and recovery of metals from aqueous streams. A special category among them are natural fibers of biological origin, which combine remarkable biosorption properties with the adaptability of useful forms for cleanup and recycling purposes. To support the efficient exploitation of these advantages, this article reviews the current state of research on the potential and real applications of natural cellulosic and protein fibers as biosorbents for the sequestration of metals from aqueous solutions. The discussion on the scientific literature reports is made in sections that consider the classification and characterization of natural fibers and the analysis of performances of lignocellulosic biofibers and wool, silk, and human hair waste fibers to the metal uptake from diluted aqueous solutions. Finally, future research directions are recommended. Compared to other reviews, this work debates, systematizes, and correlates the available data on the metal biosorption on plant and protein biofibers, under non-competitive and competitive conditions, from synthetic, simulated, and real solutions, providing a deep insight into the biosorbents based on both types of eco-friendly fibers.

## 1. Introduction

Natural cellulosic (plant) and protein (animal) fibers from biological sources (biofibers) are remarkable for their renewability, biodegradability, variety, variability, carbon dioxide neutrality, and recyclability. They fold very well on the coordinates of sustainability, which is one of the most challenging goals of today’s society. With an anticipated annual production of around 40 million tons per year by the middle of the 21st century [1], biobased fibers represent a rising class of materials for the circular and ecological economy.

Biofibers have historically been used to produce textiles, ropes, carpets, fabrics, and wires [2]. Today, natural fibers, especially plant fibers, are mainly used as reinforcement for composite materials [3,4,5,6,7,8]. Natural fiber polymer composites have outstanding development prospects in many technical fields. In addition to these applications, natural fibers are also widely used in the paper, construction, food, and pharmaceutical industries as the source of biopolymers, in biofuel and energy production, medicine, cosmetics, and agrochemistry [9,10,11,12,13,14]. However, their high potential for practical applicability still waits to be fully decrypted and exploited. In this context, the use of green fibers as biosorbents instead of conventional adsorbent materials could significantly contribute to solving the pivotal universal problem of water pollution.

Among the common methods for the removal of toxic metal ions from aqueous media, adsorption occupies a prominent position [15,16,17,18,19,20]. The achievements of the adsorption method in toxic metal cleanup are determined by the selection of the adsorbent. Therefore, the suitability of a wide range of granular or powdered materials from activated carbon, metal oxides, and synthetic resins to metal organic frameworks and highly porous nanostructures for adsorptive removal of contaminants from aqueous solutions has been extensively investigated [21,22]. The research on adsorption of harmful chemical species on biological materials (biosorption) conducted in the last decades is also in this direction [23,24,25,26,27,28,29,30,31]. The corresponding biodegradable adsorbents designated as biosorbents encompass a wide range of microorganisms, agricultural wastes, industrial wastes, and other polysaccharide materials [32,33,34,35,36,37,38,39,40,41,42,43]. As an emerging type of fibrous adsorption material, natural fibers offer advantages over all the above adsorbents, including excellent kinetic properties, higher adsorption capacity due to their low resistance to mass transfer and large external surface area, high efficiency, advanced chemical stability, higher mechanical strength, and reusability [44,45,46,47]. Besides these, the versatility of their usable forms (thread, non-woven fabric, woven in different shapes) is of particular importance for the simultaneous or sequential retention of heavy metal ions and organic pollutants from industrial effluents [48,49,50,51,52]. In addition, their ease of handling, which make them suitable for field applications, especially for rapid environmental remediation after accidents of pollution [53,54], and compatibility with marine systems [55] have special relevance.

The published reviews are either specifically devoted to biofibers and their sources for the adsorption of pollutants in water purification, or cover multifaceted applications of some natural fibers, of which those related to pollution control and remediation are briefly addressed (Table 1). Despite the large number of reviews systematized in Table 1, not a single one deals exclusively with the applicability of natural fibers for the separation of metals from industrial effluents. At the same time, the majority of the articles in Table 1 focus on reviewing the most widely investigated vegetable fibers for this purpose, while protein fibers are little explored. Moreover, heavy metals are mainly studied, while other toxic metals are rarely considered, as are the issues related to metal recovery from loaded adsorbents.

The originality and novelty of this work is that it connects, for the first time, the information on the biosorption capabilities of both cellulosic and protein natural fibers (raw, processed, and waste) for the removal and recovery of both toxic and valuable metals from diluted aqueous solutions. In addition to its integrative nature, this article is distinguished from the other reviews by the following elements: (i) focus on the ways in which the biosorption activity of natural fibers is affected by industrial competitive conditions; (ii) discussions and comparisons, from two different perspectives, namely natural plant and animal fibers as potential and practical biosorbents for metal uptake. After an overview on biofiber classification and properties, the major debated issues concern: (i) plant natural fibers for metal biosorption from mono- and multi-component aqueous synthetic solutions; (ii) protein natural fibers for metal sequestration from mono-metallic and poly-metallic synthetic solutions; and (iii) practical applications of metal biosorption on natural fibers to real solutions for decontamination and recycling goals.

Given the wide range of green fibers, this article will be inevitably incomplete, but its aim is to provide a comparative analysis on practical coordinates, that will be helpful to guide and promote sustainable approaches based on natural fibers in the separation and recovery processes of metals.

## 2. Classification and Characterization of Biofibers

### 2.1. Types of Biofibers

Biofibers, which are the predominant class of natural fibers can be described as polymeric materials of biological origin that are intact, thin, long, and easily bendable to form an elongated tissue [82]. Depending on the natural source of the fibers, they can be divided into two groups, namely plant (vegetable) fibers and animal fibers (Figure 1). Taking into account their chemical composition, biofibers can be further classified as follows: (i) lignocellulosic fibers; (ii) protein fibers.

Natural vegetable fibers originating from more than 2000 kinds of plants worldwide consist mainly of cellulose, hemicellulose, and lignin and are also known as lignocellulosic fibers [83]. As can be seen in Figure 1, there are six principal types of vegetable fibers that differ by the plant part from which they are collected: (i) bast fibers (jute, flax, hemp, kenaf, ramie, etc.); (ii) leaf fibers (sisal, agave, pineapple, banana, date palm, etc.); (iii) seed fibers (cotton, kapok, milkweed, etc.); (iv) fruit fibers (coir, oil palm, luffa, etc.); (v) grass/reed fibers (bamboo, bagasse, corn, wheat, etc.); (vi) hardwood and softwood fibers [84,85,86]. On the basis of plant usefulness, lignocellulosic fibers are categorized as primary or secondary fibers [87]. Thus, flax, hemp, jute, kenaf, or sisal belong to the primary fibers because they are obtained from plants grown only the production of biofibers [13]. The byproducts of other main plant uses, such as food, fuel [84], and feedstock [88], are the sources of secondary fibers (banana, coir, oil palm, bagasse, pineapple, etc.).

The second group of natural fibers includes animal fibers consisting of proteins (keratins). According to Figure 1, the main representatives of this category of biofibers are wool and other animal hairs (α-keratin fibers) and silk (fibroin fibers), respectively. In terms of origin, animal fibers are categorized as hair and secreted fibers [89]. Hair fibers such as wool are obtained from different species of animals (sheep, alpaca, angora, etc.) [87]. Secretions (silk fibers) are obtained from the larvae of about 14,000 species of butterflies and about 4000 species of spiders [13]. However, the best-known silk fibers are produced from the cocoon of the larvae *Bombyx mori*.

### 2.2. General Properties of Biofibers

Natural plant and animal fibers possess a variety of properties of physical, chemical, mechanical, electrical, thermal, biological, optical, acoustic, or ecological nature, which make them very attractive biomaterials [90]. The overall properties of biofibers are determined by their chemical composition and structure and are characterized by high variability and heterogeneity. Due to this variability, characterization and comparison of biofibers based on the vast amount of literature data is challenging. In this context, the properties of natural fibers, especially the mechanical properties of the cellulose-based biofibers, have been extensively covered in an impressive number of reviews [12,84,91,92,93,94,95,96] and are beyond the planned scope of this paper. Therefore, to provide an overview, they are briefly discussed below. The points relevant to the proposed objective are highlighted in the next sections.

Natural fibers are characterized by their environmental friendliness, worldwide occurrence, low cost, air permeability, low abrasiveness, tunability, and low energy consumption, and have intermediate mechanical properties and higher moisture and temperature sensitivity than synthetic fibers [85]. The main properties of biofibers are summarized in Table 2. The natural plant fibers have many similar properties, while the specificity of the proteins listed in Table 2 leads to significant differences between the characteristics of each animal fiber [59].

The presence of keratins, which are more sensitive to chemical attack and harsh environmental conditions than cellulose, weakens the properties of natural fibers of animal origin [101]. Thus, compared to the hydrophilic and polar lignocellulosic fibers, the protein-based biofibers are less hydrophilic and exhibit shorter fiber length, lower strength (with the exception of silk), and lower stiffness. However, the natural protein fibers are more bioactive, and their elasticity and elongation are significantly greater than those of plant fibers [108]. The properties of bio-based fibers can be improved and tailored to the desired application area through a wide range of physical, chemical, and biological treatments.

### 2.3. Properties of Natural Fibers with a Deciding Influence on Their Biosorption Potential

Biosorption is an environmentally friendly process of metal separation from a liquid phase as a result of its retention via physical and chemical bonds on a bio-based material (biosorbent) [24,25]. The hundreds of lignocellulosic and non-cellulosic biomaterials that have been proposed over time as metal biosorbents derived from algae [37,109], fungi [110,111], bacteria [36,112], yeasts [113], agricultural [38,39], and industrial [40] wastes.

As with all biosorbents, the most important property controlling the biosorption capability of natural fibers is their unique surface functionality with high reactivity. The uptake of metal ions on plant biofibers is due to the surface active functional groups, such as the carboxylic groups of hemicelluloses and lignin, the hydroxyl groups in cellulose, hemicelluloses, and lignin, the phenolic groups of lignin, and the carbonyl groups of hemicelluloses [114]. The biosorption efficiency of animal biofibers is related to the presence of hydrophilic surface functional groups of hydroxyl, amino, thiol, and carboxyl types in their constituent amino acids [74]. In addition to the type, content, and accessibility of surface active groups, the specific surface area, pore volume, pore size, pore size distribution, and surface charge have a great influence on the biosorption activity of biomass materials [115,116,117]. The literature review revealed the data on the surface properties of biofibers as environmentally friendly adsorbents are insufficient and inconsistent. In this context, Table 3 characterizes some natural fibers selected from the few biosorption studies that relate to these properties.

The surface properties summarized in Table 3 are in good agreement with the finding that natural fiber based biosorbents are highly competitive with others. A detailed study has established the following sequence of Pb(II) biosorptive removal efficiency: coir fibers > jute fibers > sawdust > groundnut shells [133]. Among the seven biological materials (wool, olive cake, sawdust, pine needles, almond shells, cactus leaves, and charcoal) used for the biosorption of Cr(VI), wool fibers, which could remove 81% of Cr(VI), were found to be the most efficient [134].

In addition, because of the remarkable ability of natural raw fibers to be subjected to various treatments, the improvement in the surface properties such as those listed in Table 3 can be imparted to the biofibers as well as other desirable properties for maximum biosorption performance. The treatment methods that can be used to modify the properties of biofibers have been ordered as follows: (i) physical methods; (ii) chemical methods; (iii) biological methods [59]. As can be seen from the numerous reviews on this topic, the physical and chemical methods are the most popular [2,69,74,75,107,135,136]. Physical fiber modification methods particularly aimed at removing surface impurities and increasing specific surface area and porosity may include size reduction by cutting or grinding, thermal treatments, and plasma treatments [60,98]. The high reactivity of surface functional groups on both types of biofibers allows the application of a wide range of chemical methods to develop modified natural fibers with advanced multifunctional properties of biosorption. Commonly used chemical treatments include mercerization, acetylation, dewaxing, etherification, esterification, oxidation, and graft copolymerization [59,99,115].

## 3. Biofibers for the Uptake of Metal Ions from Synthetic Aqueous Solutions

### 3.1. Metal Biosorption Approaches on Natural Fibers at Laboratory Scale

The successful use of fibers from renewable resources as biofilters for the biosorptive separation of metals depends on the extent to which they meet specific requirements expressed as high biosorption capacity, satisfactory selectivity at different concentrations, high efficiency in sequential or simultaneous removal of toxic metal ions, favorable kinetics, good stability, recyclability, and adaptability to different designs (batch and fixed bed column systems of biosorption) and environmental conditions [28,30,32].

From this point of view, batch equilibrium and kinetic studies provided most of the knowledge on the removal properties of natural fibers, such as the maximum capacity of metal biosorption and the biosorption rate. These involve immersing a suitable mass of biofibers in a known volume of a synthetic aqueous solution until equilibrium is reached, whereupon the phases are separated [137]. Figure 2 shows the reported approaches for batch biosorption of metal ions on natural fibers, ordered by the number of components number present in the aqueous solution used for the study. The discussions of the biosorption capabilities of natural fibers for metal removal and recovery in the next sections rely on this distinction. This is intended to underscore the need to fill the gap resulting from the lack of experimental data from batch biosorption studies with mixed synthetic solutions, as these are of great practical importance for real-world applications.

Additionally, for practical purposes, the information provided by fixed bed column studies is much more relevant and useful. In fixed bed biosorption methods, which can be used to the treatment of large sample volumes, the aqueous metal solution flows continuously at a defined flow rate through a column filled with a given amount of biosorbent [138]. However, there is a lack of experimental data on the use of biofibers in fixed bed column systems for metal biosorption. The studies evaluating the efficiency of continuous biosorption of Cr(VI) from synthetic aqueous solutions by short-chain polyaniline synthesized on jute fibers [139], *Hibiscus Canabicus* kenaf fibers [140], and wool fibers [141] can be mentioned as proof of concept.

### 3.2. Natural Plant Fibers for Metal Biosorption from Synthetic Aqueous Solutions

#### 3.2.1. Non-Competitive Biosorption

Like all lignocellulosic biomasses, natural plant fibers proposed as biosorbents for the metal removal from mono-component synthetic aqueous solutions often showed performances very close to those of ion exchange resins [137]. Their biosorption behavior is consistent with the Langmuir isotherm model and pseudo-second order kinetics. Besides the above mentioned physical and chemical properties of the lignocellulosic fibers, the degree of their removal efficiency also depends on the type and size of metal ions. To illustrate the multi-metal potentiality, Table 4 characterizes selected natural plant fibers based on their maximum capacity of biosorption for representative heavy metals commonly found in metal-laden wastewaters. The selection was made among works that investigated the biosorption of at least three metal ions from single solutions on plant biofibers.

Table 4 reflects the lack of studies that can serve as a reference for reliable comparisons between the biosorption properties of different types of lignocellulosic fibers, by using identical experimental conditions. Among the existing studies, it is worth mentioning that of Lee and Boswell, who comparatively evaluated the ability of coconut coir, kenaf core, kenaf bast, and cotton fibers to act as biosorptive media for Cu(II), Zn(II), and Ni(II) (Table 4) and showed that biosorption efficiency is not dependent on lignin content [148]. On the other hand, Mongiovi and coworkers compared flax- and hemp-based felts and reported lower capacities of biosorption of Al, Cd, Co, Cu, Mn, Ni, and Zn for the hemp-based felt, but the same order of removal efficiency (Pb > Cd > Zn > Ni > Co > Al > Ni > Mn) for both materials [143]. From a different perspective, Table 4 clearly shows that the modified or functionalized forms of natural plant fibers prepared by a variety of surface chemical modification methods are more relevant than the raw fibers. A large proportion of the papers reported improved biosorption of hazardous metal ions from mono-element solutions by using superior biosorbents based on lignocellulosic fibers such as flax fibers [118], jute fibers [147,156,157], ramie fibers [158], cotton fibers [159], nonwoven cotton fabric [160,161,162], loofah fiber [163], *Luffa cylindrica* fibers [164], okra fibers [165,166], or oil palm empty fruit bunch fiber [167,168] modified by graft copolymerization. The application of some suitable reaction strategies has allowed the development of modified natural plant fiber biosorbents with a dual function, which are of great interest for the sequential remediation of organic pollutants and heavy metal ions, as well as waste management. Thus, the use of pristine jute fibers for aniline removal results in a modified biosorbent with modest biosorption capacity (8.43 mg/g), which was subsequently chemically transformed by in situ polymerization of aniline on the jute fiber surface and used for the biosorption of Cd(II) and Cr(VI), with 98% and 99% efficiency, respectively [119]. The modified palm leaf sheath fibers were recommended as an eligible candidate for the treatment of dyeing wastewaters by the following procedure: carboxymethylation of palm leaf sheath fibers → multi-hydroxylation of palm leaf sheath fibers → biosorption of reactive yellow dye on modified palm leaf sheath fibers → biosorption of Cr(VI) on modified palm leaf sheath fibers loaded with reactive yellow dye with a maximum biosorption capacity of 189.48 mg/g [169].

Data analysis showed that the mechanism of the biosorption process of metal ions from mono-contaminated model solutions on untreated and modified forms of lignocellulosic fibers is complicated due to the involvement of a variety of cellulosic and non-cellulosic functional groups. At the same time, it is obvious that much research is still needed for a comprehensive understanding. Most proposed mechanisms are based on single or combined chemical interactions, such as ion exchange, electrostatic interactions, complexation, coordination/chelation, acid-base interactions, and precipitation. The dominance of ion exchange was highlighted for the mechanism of biosorption of Cu(II), Pb(II), and Zn(II) on flax fibers [142], retention of Ni(II), Zn(II), and Fe(II) on modified coir fibers [152], and the attachment of Cu(II) to palm kernel fibers [170]. The biosorption of Cd(II) on flax fibers grafted with Cd(II)-imprinted 2-pyridylthiourea [118] and coconut fibers functionalized with thiophosphoryl groups [171] was mainly explained on the basis of a chelation mechanism. The involvement of electrostatic attraction and ion exchange in the binding of Pb(II) to the untreated and chemically modified flax fibers [172] and polyaniline–kapok fiber biocomposite [173] was suggested. In another study, As(V) biosorption on a material based on jute fiber and Fe_2_O_3_ was reported to occur by electrostatic attraction and ligand exchange [120].

The strength of the bonds between the metal ions and the natural lignocellulosic fibers was also confirmed by desorption studies [120,140,146,174,175,176,177]. A selection of the works in which some fibrous biosorbents were subjected to a number of biosorption–desorption cycles ranging from 5 to 10 is listed in Table 5. In addition to the good reusability and stability of the corresponding biosorbents, the results in Table 5 also show that the regenerated plant biofibers could be used in practical applications for the removal and recovery of metals from wastewaters without the risk of the metal-loaded material becoming another source of environmental pollution.

#### 3.2.2. Competitive Biosorption

Research on the effects of the coexisting ions on the removal performance of the natural plant fibers is not very productive. The most addressed issues relate to the influence of the type, number, and initial concentration of the foreign ions on competitive biosorption. In this context, the studies published so far have mainly investigated the effects of bi-valent heavy metal ions [48,55,118,187,188,189,190,191,192,193,194,195,196,197,198,199], light metal ions [48,181,190,195,198,199,200], and anions [195,201] on the individual or simultaneous removal of toxic metals from poly- contaminated synthetic aqueous solutions under different batch experimental conditions. Some relevant results from these works are given below.

Pejic and coworkers reported efficiencies of 93.01%, 43.9%, and 43.84% for the removal of Pb(II), Cd(II), and Zn(II) from a multi-ionic solution containing Pb(II) (0.2 mmol/L), Cd(II) (0.2 mmol/L), and Zn(II) (0.2 mmol/L) by biosorption on hemp fibers modified with sodium chlorite, which showed almost the same uptake capacity for Pb(II) under competitive and non-competitive conditions [144]. It was shown that the prevalence of competition between six metal ions for the binding sites of a hemp felt coated with a maltodextrin-1,2,3,4-butane tetracarboxylic polymer changed the Cd(II) > Cu(II) ∼ Zn(II)∼ Mn(II) > Ni(II) ∼ Co(II) order established in single solutions into the Cu(II)  > Cd(II) > Zn(II) > Mn(II) > Ni(II) ∼ Co(II) order for the biosorption of metal ions from their mixed solution [187]. For the biosorption of Zn(II), Cu(II), and Pb(II) from a ternary solution of Zn(II) (0.044 mmol/L) + Pb(II) (0.042 mmol/L) + Cu(II) (0.042 mmol/L) on flax fibers, the same strong competition and the selectivity order as being of Pb > (II) > Cu(II) ≫ Zn(II) was highlighted [188]. Jute waste fabric with 63.2% lower lignin content showed the highest affinity for Ni(II) from a poly-contaminated solution of Ni(II) (20 mg/L) + Cu(II) (20 mg/L) + Zn(II) (20 mg/L) and uptake capacities of 5.872 mg/g, 4.552 mg/g, and 4.536 mg/g for Ni(II), Cu(II), and Zn(II), respectively [189]. The use of waste cotton yarn as an environmentally friendly adsorbent for Pb(II), Cd(II), Cr(III), and As(V) was proposed based on the removal efficiencies achieved for these metal ions from binary and quaternary synthetic aqueous solutions with low concentrations ranging from 250 to 1000 µg/cm^3^ [191]. The paper by Zheng et al. reported negligible influence of co-existing ions Cu(II) (100 mg/L) and Ni(II) (100 mg/L) on the potential for selective biosorption of Cr(VI) on kapok fiber combined with polyaniline (capacity of Cr(VI) biosorption of 45 mg/g) [192]. The maximum capacity of biosorption of Pb(II) shown by the chemically modified kapok fiber with the Fenton reaction decreased from 94.41 ± 7.56 mg/g in mono-metal solutions to 51.46 mg/g in multi-metal solutions containing Pb(II), Ni(II), Cu(II), Zn(II), Cd(II), and Hg(II) at a concentration of 100 mg/L for each metal [193]. Despite the inhibitory effects among the metal ions, the performance of kenaf bast fiber in removing Pb(II), Cu(II), and Zn(II) from multi-element solutions was found to be superior to that of commercial activated carbon at initial concentrations of 100 ppm [194].

The reported slight decreases in the maximum biosorption capacity of carboxyl modified jute fibers for Pb(II), Cd(II), and Cu(II) in the presence of Na(I), K(I), Ca(II), and Mg(II) ions were correlated with the increase in light metal ion concentration from 0 to 500 mg/L [190]. Gupta and coworkers showed that the biosorption of Cr(VI) on acrylic acid grafted *Ficus carica* fibers was unaffected by the presence of Na^+^, Cl^−^, and SO_4_^2−^, decreased under the condition of Ca(II) and Mg(II) coexisting cations, and increased in the presence of HCO_3_^−^ anions [195]. The negligible effects of the presence of Na(I), K(I), and Mg(II) ions on the selectivity of cotton fibers functionalized with tetratethylenepentamine and chitosan for Cu(II), Pb(II), and Cr(III) were explained by the higher affinity of the heavy metal ions to the active sites of the biosorbent [200]. The change in surface properties of loofah fibers has been assumed to be the cause of lower uptake of Cu(II) from solutions containing NO_3_^−^ and ClO_4_^−^ anions [201].

It is worth mentioning recent works confirming the selective and advanced biosorption of U(VI) on some new biosorbents based on hemp fibers under the conditions of simulated seawaters. Thus, by using polyethylenimine and guanidinyl functionalized hemp fibers, an efficiency of more than 85% was achieved in the removal of U(VI) from artificial seawater with multiple coexisting ions and low initial uranium concentration (3.05–99.03 µg/L) [198]. The near-quantitative separation of U(VI) from simulated seawater containing 3.16, 5.29, 9.73, 51.73, 98.04, and 302.55 µg U(VI)/L using hemp fibers functionalized with imidazole-4,5-dicarboxylic acid [199] was also reported.

As can be seen from the above considerations, the consonant conclusion of all the studies is that the biosorbents based on natural plant fibers have a low biosorption capacity for the coexisting ions under competitive conditions, while they retain a pronounced affinity and selectivity for the metal ions of interest for removal and recovery.

### 3.3. Natural Protein Fibers for Metal Uptake from Synthetic Aqueous Solutions

#### 3.3.1. Mono-Metal Systems of Biosorption

In contrast to lignocellulosic biofibers, information on the use of natural fibers of animal origin as biosorbents for water detoxification are scarce and mainly refers to wool, silk, and human hair. This limitation has been justified by the inferior kinetic properties and lower capacity of biosorption of protein biofibers compared to biosorbents based on polysaccharides [202]. However, these unfavorable properties go hand in hand with the superior potential for selective biosorption of metal ions under different conditions, attributed to natural protein fibers due the wide variation in the composition of the constitutive amino acids. In some works, it has been claimed that only a higher degree of selectivity could justify the real-life applications of the natural protein fibers as green adsorptive media. Accordingly, the metal selective binding capacity of wool powders was reported to be up to nine times higher than that of two commercially available cation exchange resins, namely AG MP-50 and AG 50W-X2 [203]. The same resins showed lower performance than silk biosorbents in selective uptake of Cu(II), Cd(II), and Co(II) from aqueous solutions [204]. Against this background, and given the enormous amounts of waste and unused keratinous fibers discharged each year, the search for further scientific knowledge is of paramount interest.

Most biosorption studies have been conducted on one of the aforementioned biofibers, in the form of loose fibers, powder, or non-woven fabrics. Although accurate comparative analyses are sporadic, given their practical relevance to the selection of the most suitable biosorbent, the following ascertainments should be introduced:-wool powder and oxidized wool powder performed better than waste wool fibers in biosorption of Cu(II) and Zn(II) from mono-component synthetic aqueous solutions, the optimum pH being 6 [205];-tussah silk > *Bombyx mori* silk > wool order was determined for Co(II) uptake on different untreated protein fibers [206];-the removal efficiencies of some powdered biosorbents have shown the trends: silk powder > wool powder > cashmere guard hair powder and cashmere guard hair powder > wool powder > silk powder versus Zn(II) and Cr(VI), respectively [207];-the efficiency of biosorption of Pb(II) from aqueous solutions with pH = 5.8 on human hair, goat hair and sheep wool has reached the highest value (33 mg/g) on human hair [208].

Against this background and taking into account the specifics of the properties of the individual fiber types, the metal removal and desorption properties of wool fibers, silk fibers, and human hair waste are discussed separately below.

##### Wool Biosorbents

The most targeted protein fibers are wool fibers, through studies mainly concerned with the behavior of low quality native wool, wool waste from industrial processes, and recycled woolen textiles in the biosorption of heavy metal ions (especially Cu(II), Cd(II), Zn(II), and Cr(III)) from mono-metal synthetic aqueous solutions with initial metal concentrations below 10 mmol/L. As in the case of plant biofibers, the Langmuir isotherm model and the pseudo-second order kinetic model are the optimal solutions for the description of the metal uptake on wool fibers. The significant results of these works are presented in Table 6.

The focus was to find the best treatment method to improve the biosorption properties of wool fibers. To achieve this goal, wool fibers recommended as potential materials for wastewater treatment were often modified by the irradiation with accelerated electron-beam and graft copolymerization. The significant increase in metal uptake on electron irradiated wool compared to non-irradiated wool (Table 6) is strongly dependent on the adsorbed energy dose [216,218,219,220,221,222,223,224]. Among the biosorbents based on wool fibers modified by graft-copolymerization [59], a special role is played by the amidoximated wool fibers, which have shown high efficiency and selectivity in the recovery of U(VI) from aqueous solutions [126,217,225,226]. Thus, in addition to the favorable biosorption properties shown in Table 6, the amidoxime functionalized wool fibers loaded with ZnO nanoparticles are remarkable for their antibiofouling properties, exhibited by the good inhibition of aerobic bacteria (*S. aureus* and *E. coli*), anaerobic and sulfate-reducing bacteria, and *C. albicans* fungus [217]. With a maximum capacity of U(VI) biosorption of 113.12 mg/g and attractive antibacterial properties [226], amidoxime-functionalized wool fibers with nano-TiO_2_ particles have been described as a precursor of a new class of biosorbents for recycling U(VI) from seawater.

To obtain a complete picture, the promising desorption properties of some wool biosorbents are described in Table 7.

In addition to the high desorption efficiency, mostly with EDTA, the tested wool fibers have also shown no significant loss of biosorption activity over three to six cycles. For example, the wool graft polyacrylamidoxime exhibited a 5% decrease in Hg(II) desorption efficiency at the end of the fifth cycle [228]. The level of biosorption capacity of the chelating wool fibers in Table 7 was maintained at 90% and 93%, respectively, after five cycles [229,230].

##### Biosorbents Based on Silk Fibers

Unlike other natural fibers, the metal binding capacity of silk fibers is determined by the extent of degumming treatment that must be performed prior to use [231]. For example, an increase in Cd(II) loading from 55.2% in normally degummed Eri silk fibers to 86.5% in intensively degummed Eri silk fibers has been reported [204]. Furthermore, it was shown that the uptake of Ag(I), Cu(II), and Co(II) on *Bombyx mori* silk fabrics [232] and *Bombyx mori* and *Antherae pernyi* silk fibers [206], untreated and chemically modified with tannic acid and by acylation with ethylenediaminetetracetic dianhydride, was strongly dependent on fiber weight gain. The results of an IR study have shown the dependence of the binding mode of Cu(II) and Co(II) on chemically modified *Bombyx mori* and Tussah silk on the fiber composition, the nature of metal ion, and the type of modifying agent [233]. All these studies have also demonstrated the reversibility of interactions between the silk fibers and the tested metal ions, as well as the antimicrobial properties of the metal-containing silk fabrics.

The valorization of these properties for metal ion removal and recycling applications has been hindered by the high cost of raw silk fibers, which are less economical than natural plant fibers [75,234,235]. However, in recent years, this trend has changed due to the drastic increase in the waste silk fiber amounts and the need to find new ways for their high-value utilization. For example, by using polydopamine-modified waste silk fabric, rapid removal of 100%, 99%, and 93% removal of Cd(II), Cu(II), and Ni(II) from an aqueous solution with an initial metal concentration of 60 mg/L and an almost quantitative desorption of the retained metals within 5 min were achieved [236]. With maximum biosorption capacities of 8.03 mg/g, 7.42 mg/g, and 7.49 mg/g for Cd(II), Cu(II), and Ni(II), respectively, and the ability to uptake heavy metal ions from dyeing aqueous solutions, waste silk fabric modified by tannic acid has been proposed as a viable biosorbent for industrial wastewater treatment [237].

##### Biosorbents Based on Human Hair Waste

Although very rarely investigated for potential applications in the treatment of metal-contaminated aquatic systems, discarded human hair offers some significant advantages over analogous biosorbents. Aside from its universal availability, human hair waste could be used directly as a water filter, without the need for additional pretreatments such as pelletization or immobilization. The data on the performance of human hair waste to the removal of pollutant metal cations from individual synthetic aqueous solutions are systematized in Table 8. Some authors of the papers listed in Table 8 concluded that the use of human hair waste is more economical than commercial activated carbons [238,239] and reported biosorption efficiency similar to that of sheep fur [239]. The heavy metal uptake on human hair waste has been attributed to electrostatic interactions, coordination exchange interactions, and physisorption [132,238,239]. The preliminary desorption experiments showed good recovery of the biosorption properties of human hair waste and a high percentage of the loaded metal ions desorption [240,241,242].

#### 3.3.2. Multi-Metal Systems of Biosorption

The studies on the behavior of protein biofibers in the uptake of metal ions from poly-metal aqueous synthetic solutions are by no means numerous. Those that already exist followed two lines of investigation: (i) description of the competitive effects in multi-component systems of biosorption [243,244,245,246,247]; (ii) evaluation of the industrial applicability of biosorbents based on wool fibers through studies on simulated industrial effluents containing different ions [126,134,248,249,250]. From the first category, the recent studies on the competitive biosorption of Cr(III) and Cu(II) from binary solutions on electron beam-irradiated sheep wool are of particular importance [243,244]. In one of these works, the tailoring of wool selectivity to Cr(III) and Cu(II) from bi-metal solutions with initial concentrations of each cation between 15 and 35 mmol/dm^3^ by means of the absorbed energy dose control was proposed [243]. In another work, the significant effects of competing cations on the uptake of Cr(III) and Cu(II) from binary solutions on electron beam irradiated wool were highlighted by comparing the biosorption isotherms [244]. On the other hand, the decrease in the uptake of Au(III) from multi-metal solutions on human hair waste because of the biosorption of Ag(I), Pd(II), and Pt(IV) was up to a level where the selectivity for Au(III) is still predominant [247]. From the other perspective, amidoximated wool fibers displayed a high U uptake capacity (25–35 mg/g) for biosorption of uranyl ions (102.8 mg/L) from simulated nuclear industry effluents, which also contained Cd(II) (89.05 mg/L); Ce(III) (100.1 mg/L); Eu(III) (97.95 mg/L); Co(II) (93.7 mg/L); La(III) (99.7 mg/L); Mn(II) (101.7 mg/L) and Ni(II) (85.85 mg/L); and Sm(III) (102 mg/L), at pH = 4 [126]. Chlorine-treated woven wool fabric has been able to recover 70% of Pd(II) and 65% of Au(III) from an aqueous solution containing 100 mg/L each of the following ions: Au, Pd, Pt, Ru, Ir, Hf, Sb, Sn, and Te [250].

## 4. Applications of Biosorbents Based on Natural Fibers to Metal Removal from Real Wastewaters

The available information on the compatibility of biofibers with the real systems of metal-laden wastewater treatment is summarized in Table 9. The fibrous biosorbents listed in Table 9 showed excellent performance, reflected in the removal efficiency and contact time values for the uptake of selected metal ions from diluted real solutions with complex matrices. However, the relevance of the data presented in Table 9 is significantly limited by the fact that the practicability of biosorbents based on natural fibers has been investigated in only a very small number of studies conducted under batch conditions with small volumes of wastewaters.

## 5. Future Perspectives

The expected transition of natural fibers from suitable candidates to practical biosorbents implementable in greener technologies for the removal, recycling, and reuse of metals requires extensive research focusing in particular on the following:-expanding the range of natural fibers tested for biofilter function and target metals;-complete clarification and quantification of the relationships between chemical composition, structure, and properties of the bio-based fibers;-replacement of pollutant chemical methods applied for the treatment of biofibers with cleaner procedures;-a substantial increase in the number of studies on: (i) competitive biosorption; (ii) fixed bed column biosorption; (iii) desorption–regeneration; (iv) disposal of exhausted biosorbents;-thorough deciphering of the biosorption mechanism;-expanding the process scale;-economic analyses;-strong expansion of work on real samples

## 6. Conclusions

A comprehensive analysis of the scientific literature was carried out, addressing the function of natural lignocellulosic and protein fibers as biosorbents for the removal and recovery of metals from synthetic and real aqueous solutions. Most attention has been paid to natural plant fibers, batch biosorption systems, and studies on the Cu(II), Zn(II), Cd(II), and Pb(II) ions uptake from mono-component synthetic aqueous solutions. In contrast, natural protein fibers, continuous biosorption systems, and multi-metal synthetic aqueous solutions have been much less studied. The proposed mechanisms of metal biosorption on natural fibers are dominated by interactions of ion exchange, electrostatic type, complexation, and coordination/chelation. The promising results of the desorption studies are strong evidence of the good recyclability of the biosorbents based on natural fibers, even if their number is very limited. The preliminary results of the tests with real effluents are encouraging for future industrial applications of biofibers for the severe reduction of water contamination with metal ions and the recovery of valuable metals. Research on the metal uptake capabilities of natural fibers, especially those of animal origin, needs to be continued and deepened, especially in the context of real-world scenarios and a pilot- and large-scale applications.

## Figures and Tables

**Figure 1 polymers-15-02178-f001:**
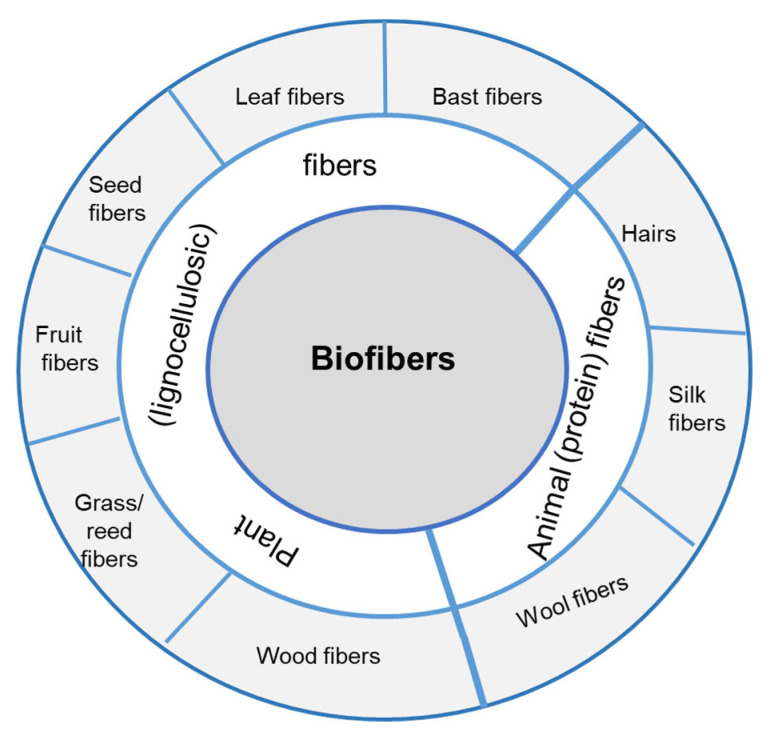
Biofiber classification.

**Figure 2 polymers-15-02178-f002:**
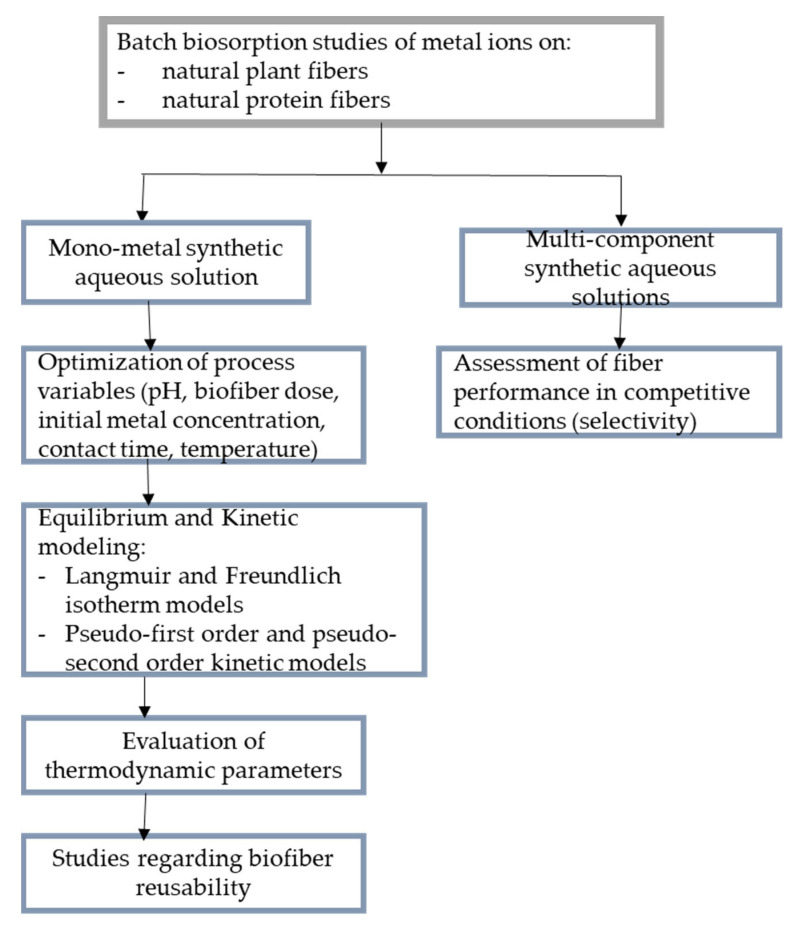
Schematic representation of the principal issues targeted in batch studies regarding the removal of metals from synthetic aqueous solutions by biosorption on natural fibers.

**Table 1 polymers-15-02178-t001:** Review articles on eco-friendly adsorbents based on natural fibers for environmental remediation.

Topic	Main Addressed Issues				Ref.
Fibrous adsorbents for wastewater treatment					
-Modified textile fibers	Fibrous ion exchangers based on fibers of cellulose, wool polyacrylonitrile, polypropylene, polyethylene terephtalate, and polyamide for remediation of heavy metal ions and dyes from aqueous effluents				[56]
-Natural and synthetic fibers	Natural and synthetic fibers for oil spill treatment and removal of trace metals and dyes; representative fibrous natural and synthetic polymer adsorbents for uranium remediation in wastewaters and sea water				[57,58]
-Natural fibers modified by graft copolymerization	Treatment for natural fibers; graft copolymerization onto natural fibers; application of grafted natural fibers for removal of heavy metal ions, dyes, other toxic pollutants; recovery of precious/ are earth metals				[59]
-Natural plant fibers	Overview of natural fibers; fiber treatment; types of physical forms of adsorbent materials; natural plant fibers for effluent treatment; adsorption ability of agro-fiber wastes for textile industrial pollutants (dyes, heavy metal ions, oils)				[60,61]
-Hemp fibers	Sorption removal of pollutants from aqueous solutions by different types of hemp fibers; the applicability of sorbents based on hemp fibers for water and wastewater treatment				[62]
-Data palm surface fiber	Types and characteristics of data palm fibers; performances of data palm surface fiber in the removal of pollutants (acid and basic dyes, heavy metals, pesticides, oils) from wastewaters				[63]
-Kapok fibers	Pretreatments and surface modification of kapok fibers; heavy metal and dye adsorption on modified kapok fibers				[64]
*-Luffa* fibers	Effective parameters in adsorption; characteristics of Luffa and its derivations; Luffa, its composites, preparation, and comparisons with other adsorbents				[65]
Adsorbents based on a certain type of natural fiber precursor biomaterial					
-Hemp	Biosorption, a useful decontamination process for contaminant removal; hemp-based materials (raw, modified, impregnated, carbonized, composite) as biosorbents of metals				[66,67,68]
-Cotton	Isotherms, kinetics, thermodynamics, and reusability of unmodified and surface-modified cotton-based adsorbents for heavy metals trapping				[69]
-Oil palm	The achievements of oil palm biomass (trunks, fronds, leaves, empty fruit bunches, shells, etc.) to the removal of dyes, pesticides, heavy metals, phenolic compounds, various gaseous pollutants				[70,71,72]
*-Luffa cylindrica*	*Luffa cylindrica*-based materials for adsorption of toxic metals, dyes, and emerging pollutant				[73]
-Keratinous materials	General characteristics of keratins; extraction of keratins and fabrication of materials; mechanism of pollutant removal; electrostatic characteristics of keratin materials; removal of oily substances; removal of metal ions; regeneration of adsorbents				[74]
-Silk	Structure and properties of silk; degumming and regeneration of silk fibroins; functionalization of silk materials; silk biomaterials for the removal of toxic ions and pollutants from effluents, oil–water separation and unidirectional water collection and transport				[75]
*Various applications of natural fibers, including environmental ones* -Hemp -Cotton wastes -Kapok -Raw wool wastes -Hair human waste	Oil spill cleanup	Removal of dyes	Removal of heavy metals	Removal of phenolic compounds	[76] [77] [78,79] [80] [81]
	x x x	x x x	x x x x x	x	

**Table 2 polymers-15-02178-t002:** Short characterization of natural plant and animal fibers [12,62,83,97,98,99,100,101,102,103,104,105,106,107].

Criterion of Characterization	Plant Fibers	Animal Fibers
Availability	Infinite	Limited
Variability	According to the species and maturity of plants, geographical location, origin, time and season of the year, quality, mode of fiber extraction, and processing	Owing to the variability of animal species and individuals and food types
Chemical composition	Cellulose consisting of repeating units of D-anhydroglucose joined by β-1,4-glycosidic linkages: 30–80%; Hemicelluloses (d-xylose, d-mannose, d-glucose, d-galactose, l-arabinose, l-rhamnose): 7–40%; Lignin (polymer of phenylpropane units with three different aromatic units): 3–33%	>95% pure proteins Wool: α-keratins with high content of cystine (sulfur containing amino acids) Silk: 72–81% fibroin rich in alanine, glycine, tyrosine, and serine amino- acids; 19–28% sericin made up of amino acids such as serine, glycine, aspartic acid, glutamic acid
Structure	Layered structure: center lumen → secondary wall (S3, S2 and S1) with the S2 middle layer comprising of microfibrils that are made up of 30–100 molecules of cellulose and helically wound → primary wall	Core shell structure: -wool: cortex (inner protein core) and cuticle (the surface shell that is composed of 3 layers -silk: inner core of protein, a protein skin, and some types of coating
Density	Low (1.35–1.7 g/cm^3^)	Low (silk: 1.3 g/cm^3^)
Moisture regain	8–13.75%	Wool: up to 15–17%; silk: >9–11%
Mechanical properties	Relatively good strength, high stiffness; Order of tensile strength and Young’s modulus: bast fibers > leaf fibers > seed fibers	Moderate strength, resiliency, and elasticity; among all natural fibers, wool has the smallest mechttensil strength, and silk has a very high rigidity
Chemical properties	Sensitive to mineral acids and resistant to alkalis	Resistant to acids, sensitive to alkalis and oxidizing agents
Thermal properties	Low value of thermal conductivity: (0.29–0.32 W/mK) Low temperature resistance (degradation can begin at 170–200 °C)	Lower thermal conductivity (wool: 0.038- 0.054 W/mK) Temperature of silk thermal decomposition > 1500 °C
Biological properties	Antimicrobial capacity	Antimicrobial activity

**Table 3 polymers-15-02178-t003:** Description of biofibers by means of some key surface properties for their biosorption capacity.

Biofiber	Specific Surface Area	Pore Volume	Other Reported Characteristics	Reference
Flax	3.25 m^2^/g		Elemental composition: C (40.3%); H (5.7%); N (0.02%); mean diameter of fibers: 24.3 µm	[118]
Jute	0.998 m^2^/g 0.57 m^2^/g 1.25 m^2^/g	0.0021 cm^3^/g	Average pore size: 6.12 nm Elemental composition: C (54.62%); O (42.71%) Elemental composition: C (46.15%); O (53.85%)	[119] [53] [120]
Cotton	15.83 m^2^/g	0.033 cm^3^/g	Micropore volume: 0.0066 cm^3^/g; mesoporous volume: 0.035 cm^3^/g	[121]
Coconut	3.6672 m^2^/g	0.00360 cm^3^/g	Surface in pores: 0.440 m^2^/g; total area in pores: 1.921 m^2^/g; Elemental analysis: humidity (3.61%); ash (2.03%); C (42.57%); H (4.53%); N (0.64%)	[122]
		3.9 cm^3^/g	pH = 5.35; cation exchange capacity: 64 mg/100 g; surface charge: 5.39 × 10^24^ meq/m^2^	[123]
Luffa cylindrica	0.966 m^2^/g	0.001 cm^3^/g	Point of zero charge (pH_PZC_) = 7.14	[124]
Wool	159 m^2^/g	7.6 × 10^−3^ cm^3^/g	Isoelectric point pH ≈ 4; average pore diameter: 1.8 nm Average tensile strength: 3.23 cN/dtex; elongation: 4.68 mm	[125] [126]
	0.67 m^2^/g		Amount of carboxylic functions: 1.7 mmol/g Elemental composition: C (60.4%); N (14.7%); O (19.1%); S (4.0%); average fiber diameter: 66.0 ± 6.7 µm	[127] [128]
Silk	15.835 m^2^/g	0.017 cm^3^/g	Elemental composition: C (49%); N (17%); O (33%) General isoelectric point around 1.2–2.8 pH values; surface elemental composition: C (70.75%); O (17.84%); N (10.94%)	[129] [130]
Human hair waste	1.36 m^2^/g		pH = 5.43; elemental composition: C (72.3%); O (26.5%); S (1.2%); point of zero charge (pH_PZC_) = 6.9363	[131] [132]

**Table 4 polymers-15-02178-t004:** Performance description of some selected lignocellulosic fibers used to treat monometallic solutions by means of maximum capacity of metal biosorption.

Biosorbent; Targeted Metals	Working Conditions			Maximum Capacity of Biosorption (mg/g)					
	pH of Solution	Biomass Dose	Contact Time	Pb	Cd	Cu	Zn	Ni	Reference
Flax fibers; Pb, Cu, Zn	4–6 (Pb, Cu) and 7 for Zn	0.5 g/250 mL of solution	60 min	10.741		9.921	8.453		[142]
Flax based felt; Al, Cd, Co, Cu, Mn, Ni, Zn	4,5,6	1 g/100 mL of solution	60 min		3.00	5.53	2.05	1.56	[143]
Waste short hemp fibers; Pb, Cd, Zn	5.5	0.5 g/200 mL of solution	2 h	16.16			5.9	4.57	[144]
Natural hemp fibers modified with ß-mercaptopropionic acid; Ag, Cd, Pb	5.75 (Cd) 3.03 (Pb)	0.5 g/25 mL of solution	24 h	22.97	14.05				[145]
Hemp based felt: Al, Cd, Co, Cu, Mn, Ni, Zn	4, 5, 6	1 g/100 mL of solution	60 min		1.02	4.51	0.76	0.53	[143]
Unmodified jute fibers; Cu, Zn, Ni	5.5	1 g/50 mL of solution	120 min			4.23	3.55	3.37	[146]
Dye loaded jute fibers; Cu, Zn, Ni	5.5	1 g/50 mL of solution	120 min			8.40	5.95	5.26	
Oxidized jute fibers; Cu, Zn, Ni	5.5	1 g/50 mL of solution	120 min			7.73	8.02	5.57	
Aminoximated jute fibers; Pb, Cu, Ni	4	0.2 g/50 mL of solution	60 min	39.9		27.6		10.1	[147]
Natural fibers of kenaf; Cu, Ni, Zn		0.5 g/10 mL of solution	24 h			0.61	0.53	0.39	[148]
Natural fibers of cotton; Cu, Ni, Zn		0.5 g/10 mL of solution	24 h			0.03	0.18	0.07	
Natural fibers of coconut coir; Cu, Ni, Zn		0.5 g/10 mL of solution	24 h			0.42	0.31	0.20	
Natural cotton fibers modified with citric acid; Cu, Zn, Cd, Pb	5			21.62	8.22	6.12	4.53		[149]
Natural cotton fibers coated by high loading of chitosan; Cu, Ni, Pb, Cd	6.5	0.1 g/25 mL of solution	24 h	101.52	15.73	24.78		7.63	[150]
Chemically oxidized kapok fibers; Pb, Cu, Cd, Zn		1 g/ 50 mL of solution	150 min	38.46	58.47	36.9	39.37		[151]
Unmodified coir fibers; Ni, Zn, Fe	6.5	1 g/50 mL of solution	120 min				1.83	2.51	[152]
Oxidized coir fibers; Ni, Zn, Fe	6.5	1 g/50 mL of solution	120 min				7.88	4.33	
Alkali treated coir fibers; Cu, Pb, Ni, Fe	6.5	1 g/50 mL of solution	120 min	29.41		9.43		8.84	[153]
Fibers of *Opuntia fuliginosa*; Zn, Pb, Cd, Mn, Cr, Fe, Cu	5	0.5 g/75 mL of solution	8 h	30.86	30.21	53.92	34.38		[154]
Fibers of *Agave angustifolia*; Zn, Pb, Cd, Mn, Cr, Fe, Cu	5	0.5 g/75 mL of solution	8 h	25.12	34.84	14.51	22.47		
Alfa grass fibers (*Stipa Tenacissima L.*); Pb, Cu, Zn	6.3	500 mg/L of solution	25 min	14.492		11.904	8.695		[155]

**Table 5 polymers-15-02178-t005:** Reusability of selected biosorbents based on plant biofibers.

Lignocellulosic Fiber-Based Biosorbent; Reference	Targeted Metal Ions	Desorption Agent	Number of Used Cycles (n)	Maximum Capacity of Biosorption, (mg/g)		Remarks
				Original	After n Cycles	
Ethylene glycol-bis(2-aminoethylether)-N,N,N′,N′-tetraacetic acid dianhydride modified ramie fiber; [158]	Cd(II) Pb(II)	0.5 M HCl	10 10	76.8 149.7	30.5 61.3	A 20 s ultrasonic treatment after HCl desorption ensured 95–99% efficiency of regeneration
Hemp-based materials; [178] -*fibers* -untreated -treated with citric acid -*shives* -untreated -treated with citric acid	Ni(II)	Aqueous solutions of pH = 2	10	158 184 145 175		Biosorption ability loss from first to last cycle: 65–30%; 69–55% 59–24% 60–43%
Polyaniline-coated sisal fibers; [179]	Pb(II)	0.1 M HCl	10	6.53		Up to 5th cycle, the desorption efficiency has been >80%
Kapok fibers modified with diethylenetriamine pentaacetic acid; [180]	Pb(II) Cd(II) Cu(II)	1 M HCl	8	310.6 163.7 101.0	>90% of the original ones	Insignificant influence on the ester bonds
Palm leaf sheath fibers loaded with Reactive Yellow 3 dye; [169]	Cr(VI)	0.1 M NaOH	7	189.48	151.98	Desorption rate: 85% after first cycle
Cotton fibers chemically modified with - aminopyridine - aminopyrazine [181]	Cr(VI)	2wt % thiourea-HCl	6	89.66 54.92	82.34 47.28	Good flexibility of cotton fibers used as substrate
Double functional polymer brush-grafted coton fibers; [182]	Cd(II)	0.1 M solution of EDTA	6	182.27	Sufficient stable	Desorption efficiency: >90%
Chelating fibers based on cotton fabrics modified by insertion of phenylthiosemicarbazide; [49]	Au(III) Pd(II) Ag(I)	0.1 N HNO_3_	5	198.21 87.43 71.14	188.10 82.36 67.44	No noticeable loss of biosorbent activity
Carboxylated bamboo fibers; [183]	Pb(II)	0.1 M HCl	5	127.1	103.4	Recovery efficiency: 96.2% (first cycle) and 88.5% after 5 cycles
Polyethyleneimine- immobilized pineapple fiber; [184]	Cu(II) Pb(II)	0.1 M HCl	5	250 160	160 80	Better reusable performances than those of alkali-treated pineapple fibers
Alkali treated pineapple fiber Polyethyleneimine-carbamate-linked pineapple fiber; [185]	Cr(VI)	0.1 M NaOH	5	133 222	<40 >100	Proposed desorption mechanism: displacement of chromate anions with hydroxyl ions
Flax fibers; [186]	U(VI)	1.5 M HNO_3_	5	27.27	21.44	Mechanism of desorption with HNO_3_: replacement of U(VI) ions fiber surface by H^+^ ions
Coconut fibers -unmodified -modified with *Saccharomyces* *cerevisiae* yeast cells; [122]	Pb(II)	1 M citric acid and 1 M acetic acid	5	64.627 84.935		The desorption degree is dependent on the nature of desorption agent

**Table 6 polymers-15-02178-t006:** Potential biosorbents based on wool fibers for metal removal from single aqueous synthetic solutions.

Keratin Fibrous Biosorptive Material	Studied Metal Ions	Maximum Capacity of Biosorption	Other Distinctive Performances	Biosorption Mechanism	Reference
Raw wool fibers	Zn(II) Cu(II)	0.0712 mg/g 0.0726 mg/g	Maximum biosorption efficiency at pH = 7: Zn(II): 95.5%; Cu(II): 94%	Mix of chemisorption and physisorption; complexation of Zn(II) and Cu(II) with amino and carboxylic acid groups	[209]
Waste wool fibers	Zn(II) Cu(II)	0.0149 mg/g 0.0212 mg/g	Maximum biosorption efficiency at pH = 7: Cu(II): 60.4%; Zn(II): 34.4%		
Pristine wool fibers	Cr(VI)	64.5 mg/g	Removal percentage >99% for a contact time at least 5 days at pH = 1.5	Cr(VI) adsorption on wool → catalytic reduction of Cr(VI) to Cr(III)	[210]
	Cu(II) Cd(II) Ni(II) Zn(II)	0.37 mol/kg 0.31 mol/kg 0.34 mol/kg 0.29 mol/kg	Affinity order: Cu^2 +^ >Ni^2+^ ~ Cd^2 +^ > Zn^2+^; time to reach equilibrium: 90 min	Chelation	[125]
Na_2_S-treated sheep wool	Cu(II) Au(III) Cu(II) Pb(II) Cd(II)	0.817 mmmol/g 0.950 mmmol/g 26.2 mg/g 42.55 mg/g 32.46 mg/g	Ability to uptake both heavy and precious metal ions; favorable kinetics Biosorption efficiency > 80% at an initial metal concentration of 10 mg/L	Cu binding through O of carboxyl groups; binding of Au via amino and thiol groups	[211] [212]
Recycled wool-based nonwoven material - untreated	Pb(II)	4.76 mg/g	No treatment is required for biosorptive properties improvement of the recycled fibers at low concentrations of Pb; significant increase of the uptake capacity by temperature rise from 20 °C to 70 °C	Chelation	[213]
- treated with chitosan		4.95 mg/g			
- treated with low-temperature air plasma		4.72 mg/g			
- treated with chitosan and low-temperature air plasma		5.00 mg/g			
Merino wool powder treated with sodium salt of dichloroisocyanuric acid	Co(II) Cu(II) Cd(II)	7.7 ± 1.2 (moles ×10^−9^)/mg 9.1 ± 0.9 (moles ×10^−9^)/mg 8.6 ± 3.2 (moles ×10^−9^)/mg	Highest biosorption at pH 6, 7, and 8 for Cu, Cd, and Co, respectively; Much faster uptake of Cu on wool powder than wool fibers	Complexation	[203]
Maleic anhydride-modified wool	Cr(III)	43.3 mg/g	Good kinetic features; maximum level of biosorption at pH = 4.5 and 40 °C	Chemical and physical interactions	[214]
Natural wool	Zn(II)	0.62 mg/g	Best biosorbent: wool physically modified with 1% chitosan solution at pH 7; 98.19% efficiency of Zn(II) removal from a solution with initial concentration of 12.5 mg/L, at pH 5 and 25 °C	Chemical bond between depronated amino groups of wool and Zn(II), with possible formation of mono- or bi- complexes	[215]
Wool physically modified with chitosan		1.53 mg/g			
Wool chemically functionalized by chitosan		0.94 mg/g			
Electron-irradiated sheep wool (applied dose: 350 kGy)	Cr(III) Cd(II) Pb(II)	2.08 mg/g 4.95 mg/g 10.15 mg/g	Order of biosorption: Pb(II) > Cd(II) > Cr(III); improvement of biosorption properties due to the residual humidity of irradiated wool	Ion exchange and complexing reactions	[216]
Non-irradiated sheep wool		By 1.87, 1.28, and 1.39 times lower for Cr, Cd, and Pb, respectively			
Amidoxime functionalized wool fibers loaded with ZnO nanoparticles	U(VI)	95.6022 mg/g	Rapid biosorption rate; high efficiency in the pH range of 6–9	Chemical adsorption	[217]

**Table 7 polymers-15-02178-t007:** Regeneration properties of biosorbents based on natural wool fibers.

Biosorbent; Reference	Metal Ion	Desorbing Agent	Desorption Conditions	Desorption Percentage	Number of Cycles
Carboxylate functionalized wool fibers; [227]	Cu(II) Pb(II)	0.1 M solution of H_2_C_2_O_4_	200 mL of desorbing solution; contact time: 24 h; 40 °C	92.4 %	6
Coarse wool graft with polyacrylamidoxime; [228]	Hg(II) Pb(II) Cd(II)	Saturated solution of EDTA	20 mL of desorption solution; 12 h contact time; 25 °C		5
Wool-grafted-poly(cyano-acetic acid α-amino-acrylic-hydrazide) chelating fibers; [229]	Hg(II) Cu(II) Cd(II)	0.1 M solution of EDTA	100 mL of EDTA; 0.2 of metal loaded wool fibers	93.3% 96.2% 93.5%	5
Wool-grafted -poly(satin acrylic hydrazone) chelating fibers; [230]	Cu(II) Hg(II) Ni(II)	0.01 M solution of EDTA		88.6% 90.4% 85.4%	5
Amidoxime functionalized wool fibers; [225]	U(VI)	0.1 M solution of EDTA	200 mL of EDTA solution contact time: 24 h; 30 °C.	95.8 %	4
Oxidized wool fibers; [128]	Cu(II) Cd(II) Pb(II)	0.1 M HCl	15 mL of HCl solution; desorption time: 1 h; 27 °C	67–88%	3

**Table 8 polymers-15-02178-t008:** Applicability of human hair waste for metal uptake from mono-component aqueous solutions.

Biosorbent	Targeted Metal Ion	Optimum Conditions: pH; Dose of Biosorbent; Contact Time; Initial Metal Concentration	Maximum Biosorption Capacity	Remarks	Reference
Human hair waste	Cr(VI)	pH = 1; 1 g/50 mL; 50 min; 20 mg/L pH = 2; 0.5 g /100 mL; 150 min; 50 mg/L	9.852 mg/g 11.64 mg/g	69% percentage of Cr(VI) removal by regenerated biomass Physisorption reaction of endothermic nature	[132] [238]
Waste of human hair	Pb(II) Cr(VI) Cd(II)	pH = 4; 0.8 g/L; 200 min; 0.48 mmol/L	0.26 mmol/g 1.48 mmol/g 0.07 mmol/g	Multi-ionic process; biosorption enthalpy: 84.5 kJ/mol	[239]
Oxidized human hair wastes	Cr(III) Cu(II) Cd(II) Pb(II)	pH = 4; 10 g/L; 30 min; 0.18 mmol/L	9.47·10^−5^ mol/g 5.57·10^−5^ mol/g 3.77·10^−5^ mol/g 3.61·10^−5^ mol/g	Percentage of Pb(II) desorption with 0.1 M solution of EDTA: 89 ± 1%; 2 reused cycles	[240]
Human hair treated with 25% ethylenediamine-N,N,N′, N′-tetraacetic acid	Sr(II)	pH = 4; 2 g/L; 24 h; 50 mg/L	17 mg/g	Electrostatic interactions; desorption percentage of 95.4% with NaOH solution at pH = 3	[241]
Human black hair waste treated with NaOH	U(VI)	pH = 4.5; 2 g/L; 2 h; 50 mg/L	62.5 mg/g	Distribution coefficient: 100.8 mL/g; U recovery of about 62% with 1 M HNO_3_	[242]

**Table 9 polymers-15-02178-t009:** Biosorptive removal of metal ions from real solutions by using natural fibers.

Real Effluent	Metal Ion of Interest	Concentration of Targeted Metal Ion in Real Sample	Biosorbent Based on Natural Fibers	Working Conditions pH; Sample Volume; Biosorbent Dose; Contact Time; Temperature	Removal Efficiency	Reference
Real uranium mine water	U(VI)	1809 µg/L	Carboxyl/ amidoxime groups modified Luffa cylindrica fibers	pH = 5; 50 mL; 0.02 g; 24 h; 25 °C	99.7%	[124]
Aluminum powder coating wastewater	Cr(VI)	100 ppm	Natural wool fibers	pH = 2; 25 mL; 16 g/L; 2 h; 30 °C	70.6%	[134]
Natural seawater	U(VI)	3.25 µg/g	Polyethylenimine and guanidyl functionalized hemp fibers	pH = 7; 50 mL; 0.02 g; 24 h;	77.53%	[198]
Composite samples of flax retting wastewaters	Zn(II) Cu(II) Pb(II)	0.420 mg/L 0.391 mg/L 0.009 mg/L	Flax processing waste	pH = 6.5; 2 g/L; 60 min; 25 °C	97.38% 98.72% ~100%	[251]
Wastewater from: -electroplating industry -wood treatment	Cu(II)	218.3 ppm 300.1 ppm	Iminodiacetic acid modified kenaf fibers	pH = 5; 20 mL; 0.1 g; 1080 min; 25 °C	96.5% 78.2%;	[252]
Eight industrial effluents from a metal-finishing factory	Cd(II) Pb(II) Cu(II) Ni(II) Fe(II) Zn(II) Co(II) Cr(III) Al(III) Mn(II)	4.7–5.1 mg/L 0.019 mg/L 0.97–5.5 mg/L 5.5–14.3 mg/L 8.7–24.5 mg/L 3.3–11.6 mg/L 3.3–11.3 mg/L 0.3–2.1 mg/L 1.1–9.6 mg/L 4.8–22.3 mg/L	Hemp based felt	pH~5; 100 mL; 60 min; 20 °C	100% ~100% ~100% ~100% 85% 84% 69% 68% 43% 26%	[253]
Wastewater from electroplating industry Wastewater from wood treatment	Cu(II)	218.3 ppm 300.1 ppm	Phosphoric acid modified hemp fibers	pH = 5; 20 mL; 0.1 g; 1080 min; 25 °C	88.2% 61.5%	[254]
Polluted river waters	Pb(II) Cr(VI) Ni(II) Cu(II)	0.025 mg/L 0.0205 mg/L 0.0085 mg/L 0.016 mg/L	Powdered palm fruit fibers	pH = 2–4; 50 mL; 2 g; 60–80 min	73% 78% 87% 82%	[255]
Wastewater sample collected from a wastewater treatment plant	Pb(II)	200 mg/L	Natural palm tree waste fibers	pH = 6; 25 mL; 0.3 g; 120 min	99.92%	[256]
Domestic sewage effluent	Cu(II) Pb(II) Zn(II)	73.21 ppm 48.53 ppm 42.72 ppm	Imidazole- functionalized polymer graft banana fiber	pH = 3.25; 50 mL; 0.05 g; 1 h	100%	[257]

## Data Availability

Not applicable.

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
