# Peer review of "Insights into the Applications of Natural Fibers to Metal Separation from Aqueous Solutions"

_polymers, 2023, doi:10.3390/polym15092178_

Round 1

Reviewer 1 Report

Dear author(s), 

Thank you very much for inviting me to review the manuscript Insights into the applications of natural fibers to metal separation from aqueous solutions”. The manuscript is very interesting, however, there are some issues that can be immediately addressed to improve the overall impact: 

  • The novelty of the work must be clearly addressed and discussed, compare your research with existing research findings and highlight novelty. 
  • Justify the urgency of its investigation from industrial point of view.  
  • Do the authors consider the topic original or relevant in the field? Does it address a specific gap in the field? 
  • The main objective of the work must be written on the more clear and more concise way at the end of introduction section. 
  • Introduction section must be written on more quality way, i.e. more up-to-date references addressed. Research gap should be delivered on more clear way with directed necessity for the conducted research work. 
  •  Authors must establish a connection between state of the art and your paper's objectives in the introduction. Please follow the literature review with a concise and clear analysis of the state of the art, particularly the technical part. 
  • The authors of the manuscript should add few sentences in the Introduction section about the different biosorbents investigated to remove heavy metals from wastewater (you can refer to: https://doi.org/10.1080/15567036.2022.2080891) 
  • What is the way forward for this study? 
  • Authors have to remove all the typo and grammatical error in the paper which may interfere with the reading and understanding of the paper. It would be convenient for the manuscript to be reviewed by a native English speaker so that any grammatical errors may be identified are corrected. 

Reviewer 2 Report

Overall, very good and useful work. Minor objections are given in the PDF document.

Major review is not so much because of what is written, but more because of how it is written. In the text, a poetic way of writing is used at some places instead of a technical one, which can cause problems for those who are not from the English-speaking area, so it would be good if the author corrected it. It is also desirable to find some other person, instead of the manuscript author, to read and correct some uncommon and hard to read syntax constructions of sentences that sometimes make reading of text difficult (this is common problem for the review which are compilation of many documents). There are also a lot of technical deficiencies in the tables related to aligning materials in certain rows of the tables that need to be fixed in order to make the presented data clearer. All these corrections are not of essential importance for the quality of the work, but above all of technical importance in order to make the work better and easier for the reader to follow.

It would also be very good if the author could, after each table and review of the results for a certain type of fiber, give some of his own minor comments (one paragraph or up to 2-3 sentences) in the form of important observations and conclusions that can be drawn from the data shown in the table, as this would significantly improve the quality of the work and enabled readers to quickly get to the necessary data and trends, which should be the main goal of review papers.

After these corrections, the work should be accepted.

Round 2

Reviewer 1 Report

This reviewer commends the authors efforts in addressing the comments. The responses are satisfactory, and the manuscript is hereby recommended for publication.

Author Response

Thank for time and consideration

Reviewer 2 Report

Manuscript is corrected and in some cases overcorrected (author have corrected even parts that did not need to be corrected but this is mine mistake since I did not mentioned which phrases need to be more technical and less poetic and I apologize for this increased amount of work).

However, there is still an important issue that need to be corrected - Tables. Since this work is review and its main contribution are Tables they absolutely need to be clear. However, in most cases they are bad organized so it is difficult or impossible to follow the content because rows are not aligned in the same manner. And each column need to be left aligned or justified while rows need to be centered. In addition in some cases where there are two or three cases for the same type of adsorbent they need to be clearly organized in order to easy see which data belong to which case. I hope this will not discourage the author because this suggestion is in order to improve the quality of the manuscript. Some of the problems with rows and columns are listed in PDF but when I saw that there are to many of them I added comment before Table 1 to correct all issues so please organize the Tables content in order to be clear and readable. 

Author Response

I remade all tables. All changes made in the manuscript  have marked in yellow.

Thank for time and consideration